# Drivers of Radioresistance in Prostate Cancer

**DOI:** 10.3390/jcm11195637

**Published:** 2022-09-24

**Authors:** Liam King, Nijole Bernaitis, David Christie, Russ Chess-Williams, Donna Sellers, Catherine McDermott, Wendy Dare, Shailendra Anoopkumar-Dukie

**Affiliations:** 1School of Pharmacy and Medical Sciences, Griffith University, Gold Coast, QLD 4215, Australia or; 2Ramsay Pharmacy Group, Melbourne, VIC 3004, Australia; 3GenesisCare, Gold Coast, QLD 4224, Australia; 4Faculty of Health Sciences & Medicine, Bond University, Gold Coast, QLD 4229, Australia

**Keywords:** prostate cancer, radiotherapy, treatment resistance

## Abstract

Prostate cancer (PCa) is the second most commonly diagnosed cancer worldwide. Radiotherapy remains one of the first-line treatments in localised disease and may be used as monotherapy or in combination with other treatments such as androgen deprivation therapy or radical prostatectomy. Despite advancements in delivery methods and techniques, radiotherapy has been unable to totally overcome radioresistance resulting in treatment failure or recurrence of previously treated PCa. Various factors have been linked to the development of tumour radioresistance including abnormal tumour vasculature, oxygen depletion, glucose and energy deprivation, changes in gene expression and proteome alterations. Understanding the biological mechanisms behind radioresistance is essential in the development of therapies that are able to produce both initial and sustained response to radiotherapy. This review will investigate the different biological mechanisms utilised by PCa tumours to drive radioresistance.

## 1. Introduction

Prostate cancer (PCa) is the second most commonly diagnosed male cancer and the fourth most common cause of cancer related death in men worldwide [1]. In Australia the impact of PCa is even higher being the most commonly diagnosed male cancer and second leading cause of cancer related death in men [2]. Current treatments used for localised disease or regional spread include androgen deprivation therapy, surgical intervention (radical prostatectomy) and radiotherapy. These treatments may be used as monotherapy or combined for higher risk disease [3].

Radiotherapy can be delivered externally using linear accelerators, known as external beam radiation therapy (EBRT) or internally via the implantation of radioactive seeds into the prostate that release radiation slowly, known as brachytherapy. It utilises ionising radiation (IR) that causes DNA single and double strand breaks which prevents cell division and proliferation leading to cell death [4]. IR causes cellular damage through direct exposure, indirect DNA damage and delayed responses [5]. Indirect DNA damage occurs as a result of IR producing reactive oxygen species (ROS) via water hydrolysis [6]. 

Following the production of ROS by IR, alterations occur in a number of intracellular processes in response to the initial oxidative damage [7]. Alterations of these processes aim to promote cell survival while maintaining genetic integrity, with these processes being a part of two distinct types of pathways, the DNA damage cell cycle checkpoint pathways and DNA repair pathways [8].

At present, treatment with radiotherapy for intermediate/high risk localised PCa results in an 84% 5-year survival rate [9]. Despite this high 5-year survival treatment resistance has still been observed. Advancements in radiotherapy delivery, such as volumetric modulated arc therapy (VMAT), and dose-escalation, have improved treatment outcomes but have been unable to totally overcome the development of radioresistance in PCa tumours [10,11]. Various factors have been linked to the development of tumour radioresistance including abnormal tumour vasculature, oxygen depletion, glucose and energy deprivation, changes in gene expression and proteome alterations [12]. This review will investigate the different biological mechanisms utilised by PCa tumours to drive radioresistance.

A search of Pubmed/Medline, Embase and Cochrane Library of Systematic Reviews was conducted. Key words used in the initial search included radioresistance, prostate cancer, radiotherapy, radiosensitivity and ionising radiation. In addition, if a mechanism was identified in the initial search it was included as a key term in a secondary database search. Studies selected for inclusion into the review directly investigated the impacts of alterations of a biological mechanism on PCa response to radiotherapy. Due to the lack of clinical studies investigating the outcomes of pathway inhibition and radiotherapy response in PCa, studies focusing on any type of cancer were included in the clinical section of the review.

## 2. Androgen Receptors

The role of androgen receptors (AR) in the development, growth, survival and progression of PCa is well documented, with androgen deprivation therapy being one of the key treatments in castrate sensitive PCa [13]. ARs also play a crucial role in castrate resistant PCa which has the ability to respond to castration levels of androgens through various mechanisms including mutation and overexpression of ARs [14]. Second-generation antiandrogens including enzalutamide and abiraterone have demonstrated effectiveness in castrate resistant PCa and are used as treatments either before or after other systemic treatments such as taxane chemotherapy or lutetium radio-isotope therapy [15].

Along with their role in the growth and survival of PCa, ARs have also been found to promote radioresistance. Following a number of preliminary experiments that identified the ability of ARs to increase the expression of various DNA repair genes, Polkinghorn et. el. turned their attention to how androgens and ARs influence responses to radiotherapy. They found that LNCaP cells pre-treated with androgen for two days prior to exposure to IR demonstrated enhanced DNA repair and decreased DNA damage. To confirm the role of the AR in the observed increase in survival, they then exposed LNCaP cells to the antiandrogen ARN-509 and found that antiandrogen treatment promoted cell death as a result of increased DNA damage and decreased clonogenic survival [16]. 

Not only do ARs promote radioresistance but they have also been found to be upregulated in response to DNA damage induced by IR. Goodwin et al. concluded that IR induced upregulation of AR activity promotes expression of genes governing DNA repair, identifying DNA-dependent protein kinase catalytic subunit (DNAPKcs) as a key target of ARs following DNA damage. Furthermore, they found that ARs promote resolution of double strand breaks and resistance to DNA damage following IR exposure in vitro in LNCaP cultures, and in vivo using LNCaP mouse xenografts [17].

In addition to the effects of ARs on DNA repair pathways, it has also been hypothesised that ARs can reduce the effects of ROS produced by IR through an increase in antioxidant pathways (Figure 1). Pinthus et al. showed that exposure to testosterone resulted in increased basal ROS, which consequently increased basal levels of oxidative stress response mediators including the antioxidant enzymes catalase and manganese superoxide dismutase (MnSOD). When exposed to IR, PC-3 cells pre-treated with testosterone demonstrated greater survival than those not treated and importantly this effect was reversed by bicalutamide, an AR blocker, suggesting that androgen induced radioresistance was a result of the direct activation of ARs [18].

Previous pre-clinical work investigated the ability of the second-generation AR antagonist enzalutamide to induce radiosensitivity in LNCaP and C4-2 PCa cells grown media with or without testosterone supplementation. Ghashghaei et al. exposed these cell lines to enzalutamide and radiotherapy and conducted gene expression and survival analysis. They concluded that enzalutamide treatment enhanced the effect of radiotherapy through immune and inflammation-related pathways in LNCaP cells and metabolic pathways in C4-2 cells [19].

## 3. Ataxia Telangiectasia Mutated Gene

The ataxia telangiectasia mutated gene (ATM) is a regulator of DNA damage checkpoint response pathways which ensures genetic integrity by arresting the cell cycle to facilitate repair. Increased ATM has been found to correlate with enhanced radioresistance. Hoy et al. demonstrated the ability of an ATM inhibitor, KU-55933, to reverse microRNA-106a, an oncogenic microRNA, induced radioresistance in PCa cells (PC-3 and DU145) [20]. Previous to this study, Yan et al. used the microRNA-101 to target ATM along with DNA-PKcs to produce radiosensitivity in cancer cells [21]. While this study was not conducted in a PCa model it does demonstrate a link between ATM and cancer radioresistance. The oncogene tip60 has also be found to alter the activity of ATM and promote radioresistance. In tip60 knockdown PC-3 and DU145 cells acetylation of ATM was reduced and resulted in increased radiosensitivity compared to the parental cells [22]. 

## 4. PI3K/Akt/mTOR

The phosphatidylinositol 3-kinase (PI3K)/protein kinase-B (Akt) pathway is involved in the promotion of cell survival, proliferation and progression and if not appropriately regulated it can drive the development of cancer, including PCa. PI3Ks are a group of enzymes that phosphorylate the 3′ hydroxyl group of phosphatidylinositols [23]. This phosphorylation results in the recruitment of Akt to the cell membrane which then activates a number of downstream targets, including mammalian target of rapamycin (mTOR), that are involved in cell survival, proliferation, cycle progression, migration and angiogenesis (Figure 1) [24].

Uncontrolled activation of this pathway has not only been linked to the development of cancer but also resistance of tumours to radiotherapy. Chang et al. demonstrated an increase in activation of the PI3K/Akt/mTOR pathway in radioresistant PCa using novel radioresistant prostate cancer cell lines (PC-3RR, DU145RR and LNCaPRR) [25]. Following this, they further investigated the effects of dual inhibition of PI3K and its downstream target, mTOR. Dual inhibition resulted in a significant reduction in radioresistance by repressing colony formation, increasing apoptosis, arrest of the G2/M phase, increasing double strand breaks and reducing inactivation cell cycle checkpoint and autophagy. They also found that dual inhibition was more efficacious than inhibition of PI3K or mTOR alone [26]. Radiosensitisation as a result of dual PI3K and mTOR inhibition was also demonstrated by Potiron et. al., who showed that the novel PI3K/mTOR inhibitor BEZ235 enhanced the efficacy of radiotherapy on PC-3 and DU145 cells in vitro and tumour xenografts in mice [27]. In 2018, Chen et al. demonstrated the ability of antrocin, a component of antrodia cinnamomea, to produce radiosensitivity in PCa cells and tumour xenografts. They concluded that this occurred through the downregulation of the PI3K/Akt and MAPK pathways [28]. 

Furthermore, PI3K regulates expression of hypoxia-inducible factor-1α (HIF-1α), which then promotes growth and progression of hypoxic tumours primarily through increasing vascularisation and glucose metabolism [29].

## 5. Epidermal Growth Factor Receptor

An important upstream, activator of the PI3K/Akt pathway is the epidermal growth factor receptor (EGFR). EGFR is a cell surface receptor that plays a critical role in cell proliferation, survival, growth and inhibition of apoptosis and when over-expressed EGFR supports cancer progression [30]. 

Early studies have linked EGFR over-expression to radioresistance and demonstrated that EGFR inhibition in combination with IR leads to decreased tumour cell growth [31]. Not only is EFGR over expressed in radioresistant cells, exposure of PCa cells (PC-3, LNCaP and DU145) to IR has been found to result in an amplification of EGFR expression suggesting that this amplification may act as a survival mechanism [32].

In efforts to elucidate the mechanism behind EGFR induced radioresistance, Rajput et al. investigated the effects of shRNA-mediated EFGR knockdown in radioresistant PC-3 and DU145 cells. The study concluded that EGFR radioresistance was a result of the decreased expression of the downstream protein Rad51 with Rad51 expression being mediated by the PI3K/Akt and Erk1/2 pathways [33]. 

## 6. Nuclear Factor Kappa B

Nuclear factor kappa B (NF-κB) proteins are involved in regulating a number of pathological processes and are considered as regulators of cell homeostasis [34]. Importantly NF-κB is involved in the regulation of cell proliferation and death, and alterations in its expression can lead to the development of cancer [35]. More specifically, increased NF-κB expression can lead to cancerous transformation of normal prostate cells and drive metastatic spread of PCa tumours [35,36]. NF-κB is activated by pro-inflammatory cytokines including IL-1β and TNF-α, both of which are elevated in response to IR [37,38].

The NF-κB family consists of five proteins, RelA (p65), RelB, c-Rel, NF-κB1 (p105–p50) and NF-κB2 (p100–p52) [39]. Whilst all five of these proteins are involved in cell regulation, the literature has suggested links between ReIB expression and PCa severity, with Lessard et al. having demonstrated a significant correlation between ReIB levels and Gleeson score in PCa tissue samples [40]. Furthermore, increases in ReIB expression promotes radioresistance in PCa cell lines and inhibition of ReIB results in an increase in PCa cell death in response to IR [41,42]. It has been suggested that this radioresistance is a result of ReIB-mediated induction of MnSOD resulting in protection of the cancer cells from ROS produced by IR [43]. Furthermore, radiosensitisation of PC-3 cells has been achieved through MnSOD inhibition with microRNA, confirming the role of this enzyme in radioresistance [44].

NF-κB activity, along with many other transcription factors, is upregulated in response to IR [45]. This increase in NF-κB activity has been shown to promote radioresistance and with inhibition of NF-κB resulting in radiosensitisation of PC-3 cells. Raffoul et. al., concluded that NF-κB inhibition with genistein, an isoflavone with anticancer properties, increased PC-3 cell death through cell cycle arrest and the promotion of apoptosis [46].

## 7. Bcl-2

The Bcl-2 family is a group of pro-survival proteins whose primary role is to regulate the intrinsic apoptotic pathway [47]. Overexpression of Bcl-2 has been linked to PCa progression and upregulation appears to occur via the PI3K/NF-κB pathway [48]. Past research has suggested that Bcl-2 overexpression may also act as a prognostic marker of poor response of PCa tumours to radiotherapy [49]. A significant increase in Bcl-2 expression has been found in PCa tumours that progressed following radiotherapy compared to those where radical prostatectomy was used as the primary treatment, demonstrating the potential of elevated Bcl-2 to act as a marker of radioresistance [50]. 

Multiple studies have investigated the ability of Bcl-2 inhibition to overcome radioresistance. Firstly, a ribonucleotide reductase inhibitor, didox, has been shown to overcome radiation induced Bcl-2 expression leading to radiosensitisation in PC-3 cells [51]. Following this, An et al. investigated the effects of HA14-1, a novel Bcl-2 inhibitor, on the radiosensitivity of LnCaP and PC-3 cells. They found that treatment of these cells with HA14-1 prior to IR exposure resulted in significantly increased cell death, demonstrating that inhibition of Bcl-2 leads to radiosensitisation [52]. Furthermore, the pro-apoptotic plant stress hormone, methyl jasmonate, has been shown to produce radiosensitivity in PC-3 cells via Bcl-2 suppression [53].

Finally, downregulation of Bcl-2 in PCa tumours has been found to produce radiosensitivity in vivo. Anai et.al. demonstrated that downregulation of Bcl-2 with antisense deoxynucleotide produced in PC-3 xenografts resulted in significantly increased tumour size reduction compared to control tumours [54].

## 8. Hypoxia-Inducible Factor

Many human cancers, including PCa, exist within a hypoxic microenvironment. This hypoxic microenvironment can promote resistance to IR, believed to be due to the lack of oxygen limiting the development of ROS and ultimately limiting the indirect actions of radiation on tumour DNA [55]. 

Hypoxia-inducible factor 1α (HIF-1α) transcription is activated in hypoxic environments and expression further increases following cell exposure to IR [56]. This suggests HIF-1α plays a protective role following IR induced cell damage. HIF-1α activation leads to the upregulation of a number of genes involved in cancer cell survival and growth, including vascular endothelial growth factor (VEGF) [30]. Furthermore, expression of HIF-1α is regulated by PI3K which, as previously discussed, has been linked to cancer development and progression as well as tumour radioresistance [29]. In PCa androgens upregulate HIF-1, with dihydrotestoterone stimulating HIF-1α expression and transcriptional activity and VEGF production [57].

While extensive research has been conducted into the role of HIF-1 in cancer radioresistance, there is limited data on its role in PCa specifically. Luo et. al., investigated the effects of HIF-1α and its downstream regulator of embryogenesis and tumour progression β-catenin. They found that enhanced β-catenin nuclear translocation, induced by HIF-1α, resulted in increased cell proliferation and invasion and decreased apoptosis following irradiation [58]. In addition, elevated levels of HIF-1α in PCa tumour biopsies have been found to correlate with reduced time to biochemical failure in men following radiotherapy suggesting increased treatment resistance. However, whether HIF-1α inhibition overcame this treatment resistance was not investigated [59]. 

## 9. miR-191

miR-191 is a known oncogenic mRNA and has been found to be upregulated in several cancers including breast, colon, lung, pancreas and stomach [60,61]. Furthermore, previous research has suggested that miR-191 expression may be induced in response to IR [62]. 

In 2019 Ray et al. conducted a study investigating the role of miR-191 in PCa and resistance to radiotherapy. Firstly, through analysis of The Cancer Genome Atlas (TCGA) prostate dataset they identified that miR-191 was elevated in PCa compared to normal prostate tissue and its abundance was significantly increased with increasing Gleason score. In addition, they demonstrated that miR-191 promoted radioresistance in prostate cancer cell lines and suggested this occurred through the ability of miR-191 to target retinoid X receptor alpha (RXRA) [63].

## 10. p53

p53 is a transcription factor that induces cell cycle arrest and apoptosis in response to DNA damage, hypoxia, oncogene activation, cell adhesion and redox stress [64]. It inhibits cell proliferation and growth through regulation of gene expression of a variety of mediators and the pattern of p53 induced gene expression varies depending on the type of stress experienced by the cell [65]. Key influences of p53 on gene targets include the enhancement of BAX, a pro-apoptotic member of the Bcl-2 family, and inhibition of Nf-κB, mTOR, HIF-1α and VEGF [66,67].

Early research into the role of p53 and radioresistance suggested that clinical failure of radiotherapy is more common in tumours with abnormal p53 reactivity [68]. However, these findings were contradictory to another study that found no correlation between p53 expression and cancer-free survival [69]. Further investigations by Ritter et. al., found that increased p53 expression correlated with biochemical failure at 5-years post radiotherapy [70].

In vitro studies investigating the connection between p53 and radiotherapy are similarly conflicting. Sasaki et. al., demonstrated that upregulation of p53, using an adenovirus vector containing the wild-type p53 gene, resulted in synergistic effects increasing PC-3 and DU-145 cell death following exposure to IR [71]. In contrast, Scott et. al., demonstrated that the presence of wild-type p53 increased survival and reduced the effectiveness of radiotherapy in PC-3 cells that had p53 function restored [72]. 

## 11. Clinical Application of Pathway Inhibitors

Limited human studies investigating the outcomes of pathway inhibition on PCa radiotherapy were identified in this review, with androgen inhibition being the only mechanism having been studied in this cohort of patients.

A large prospective study conducted by Jones et al. investigated the use of short-term ADT before and during radiotherapy in men with early stage localised PCa. They found a significant difference in 10-year overall survival and two-year recurrence free survival between men treated with ADT for four months prior to and during radiotherapy compared to those that received radiotherapy alone. The authors concluded that short term ADT prior to radiotherapy was associated with decreased disease specific mortality and increased overall survival, with a post hoc risk analysis suggesting that the benefit was mainly seen in intermediate-risk but not low-risk disease [73].

Similar to Jones et.al, an additional study conducted by Denham et.al. investigated the use of three and six months of neoadjuvant ADT in men undergoing radiotherapy for locally advanced PCa. Both three and six months of ADT prior to radiotherapy resulted in significant improvements in PSA progression, local progression and event-free survival. However, significant improvement in distant progression, PCa specific mortality and all-cause mortality was only observed with six-months of neoadjuvant therapy. The study concluded that six months of neoadjuvant ADT prior to radiotherapy is an effective treatment for locally advanced PCa [74].

Consistent with the effects seen with neoadjuvant therapy further research has also demonstrated that long-term (24 months) of adjuvant ADT following radiotherapy improves biochemical control and overall survival [75].

In addition to conventional ADT, new antiandrogens such as abiraterone and enzalutamide have been investigated as adjuncts to radiotherapy. Firstly, Cho et.al conducted a phase II study in men with localised PCa treated with six months of neoadjuvant and concurrent abiraterone with ADT and radiotherapy. Following this, Kaplan et al. conducted a similar phase II study in men with intermediate risk localised PCa treated with 6 months of enzalutamide commencing 2 months prior to radiotherapy. Both studies concluded that the addition of either abiraterone or enzalutamide was safe and effective in achieving androgen suppression when used alongside radiotherapy. Further, studies including phase III randomised controlled trials are needed to confirm the role of these agents in the treatment of localised PCa with radiotherapy [76,77].

While no clinical studies were found that investigated inhibition of the other drivers of radioresistance in PCa, outcomes in other cancers have been previously studied.

The PI3K/Akt/mTOR pathway involves multiple mediators that have been investigated as potential targets for overcoming radioresistance in a clinical setting. Firstly, the PI3K inhibitor buparlisib has been shown to reduce hypoxia in human non-small cell lung cancer tumours. This finding presents buparlisib as a potential radiosensitizer, with the authors recommending further investigations into its role in as an adjunct to radiotherapy [78]. Furthermore, a preliminary clinical study conducted by Hill et al. investigated the combination of the Akt inhibitor nelfinavir and radiotherapy in patients with rectal cancer. While only having a small sample size (*n* = 10), the study demonstrated that nelfinavir-radiotherapy combination was well tolerated and increased blood flow to tumours, warranting further studies into concurrent Akt inhibition and radiotherapy [79]. However, a phase 2 study investigated the effects of mTOR inhibition with everolimus in patients being treated with chemoradiation for glioblastoma and found that the addition of everolimus did not improve progression free survival and increased the incidence of treatment related toxicities [80,81]. These studies suggest that targeting of the early mediators in this pathway, i.e., PI3K and Akt, may be a more effective approach to overcoming cancer radioresistance.

Perhaps the most widely studied group of radiosensitising agents are the EGFR inhbitors. In a clinical setting EGFR inhibition is primarily achieved through the use of tyrosine kinase inhibitors (TKIs) or with EFGR targeting antibodies, e.g., cetuximab and panitumumab. Concurrent treatment with erlotinib or gefitinib, EGFR targeting TKIs, and radiotherapy has been shown to be effective in improving progression free survival as first-line treatment of stage IV non-small cell lung cancer [82]. In addition, when used as a radiosensitiser cetuximab has been shown to improve locoregional control and reduce mortality compared to radiotherapy alone in squamous-cell carcinoma of the head and neck [83]. However, recent studies have suggested that the use of EGFR antibodies during cisplatin based chemoradiation for squamous-cell carcinoma for the head and neck does not confer survival benefit and increases the incidence of adverse events [84,85]. Furthermore, two phase II studies investigating the radiosensitising ability of panitumumab in locally advanced rectal cancer were unable to reach their primary efficacy endpoints and therefore the investigators could not recommend the use of panitumumab as a radiosensitiser in this population [86,87].

Many agents that inhibit the discussed biological mechanisms are approved for use in other diseases (Table 1). Further research is warranted specific to their potential role in PCa given they inhibit biological mechanisms identified to promote radioresistance PCa. These drugs have the potential to be repurposed allowing for more expedited research and accelerated uptake into the clinic.

## 12. Conclusions

With PCa being the second most commonly diagnosed cancer and the fourth most common cause of cancer-related death, further research is needed to determine the cause of treatment resistance and failure. Along with radical prostatectomy and androgen deprivation, radiotherapy is utilised as a first line treatment in PCa, however tumours can develop radioresistance resulting in treatment failure or disease recurrence. This review demonstrates that PCa radioresistance is complex and involves various mechanisms that regulate the cell cycle, apoptosis, DNA damage repair, tumour vasculature and antioxidant production.

A key cellular pathway that appears to be heavily involved in PCa radioresistance is the PI3K/Akt/mTOR pathway. Not only does PI3K upregulation occur in radioresistant PCa cells [24], but PI3K inhibitors also induce radiosensitivity [26,27,28]. Furthermore, this review outlined the involvement of several mediators in PCa radiosensitivity that are regulated by the PI3K/Akt/mTOR pathway, including EGFR, NF-κB, Bcl-2 and HIF-1α. 

An increase in antioxidant expression also appears to be a key mechanism by which PCa cells develop radioresistance. Both the AR and NF-κB have been implicated in the promotion of antioxidant mediated radioresistance via an increase in MnSOD activity. Furthermore, research has suggested cross-talk between a number of drivers of radioresistance including HIF-1, Nf-κB and ATM and the transcription factor NF-E2-related factor (NRF2) which regulates the expression of numerous antioxidant proteins [88]. The NRF2 pathway has also been shown to be regulated by mTOR and it has been suggested that mTOR inhibition could lead to a downregulated antioxidant response and promote radiosensitivity [89]. which again implicated the PI3K/Akt/mTOR pathway as key player in cancer radioresistance.

Despite promising pre-clinical data, clinical studies on the impacts of these pathways in PCa radioresistance are limited, with no clinical trials investigating the ability of pathway inhibitors to improve response to PCa radiotherapy. However, promising outcomes from clinical trials in other cancers and the commercial availability of inhibitors of several of the discussed pathways indicates the need for further research in this area.

In conclusion, the development of radioresistance in PCa is complex and involves various cellular mechanisms. A potential way to overcome this resistance is to use adjuvant treatments that inhibit the mechanisms utilised by PCa tumours that promote survival in response to IR exposure. Greater understanding of these mechanisms may potentially present additional targets for adjuvant treatments during radiotherapy. These treatments may enhance tumour radiosensitivity and ultimately improve therapeutic outcomes and with approved inhibitors of some of these mechanisms being commercially available, drug repurposing may present a cost-effective solution to overcoming radioresistance.

## Figures and Tables

**Figure 1 jcm-11-05637-f001:**
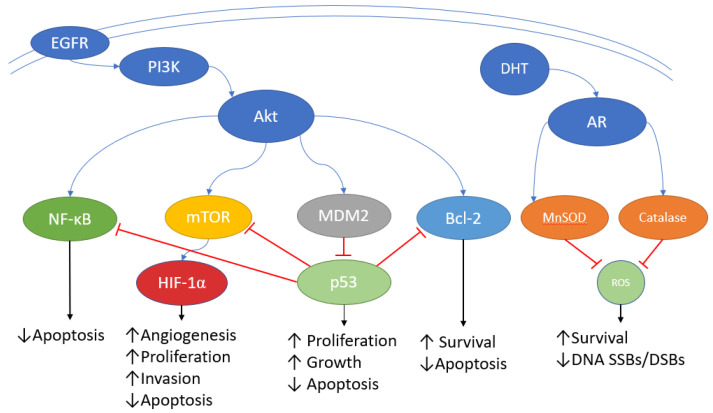
Pathways involved in the promotion of radioresistance in PCa. Overexpression of growth factor receptors such as the epidermal growth factor receptor (EGFR) promotes phosphatidylinositol 3-kinase (PI3K) activation which then activates protein kinase-B (Akt). Akt leads to increased expression nuclear factor kappa B (NF-κB), mammalian target of rapamycin (mTOR) and Bcl-2 which are all linked to radiosensitivity. Akt also increases expression of MDM2 which is a negative regulator of p53 activity. Androgen receptor (AR) activation by dihydrotestosterone (DHT) promotes activity of the antioxidants, manganese superoxide dismutase (MnSOD) and catalase, resulting in decreased reactive oxygen species (ROS) levels promoting radioresistance through reduced DNA double strand (DSB) and single strand breaks (SSB).

**Table 1 jcm-11-05637-t001:** Therapeutic Goods Administration (TGA) approved inhibitors of biological mechanisms that promote radioresistance in PCa. (mPCa = Metastaic PCa; CRPCa = castrate resistant PCa; nmCRPCa = non-metastatic castrate resistant PCa; CSPCa = castrate sensitive PCa; mCSPCa = metastatic castrate sensitive PCa).

Target	Inhibitor	Approved Indications
AR	Abiraterone	mPCa
	Enzalutamide	CRPCa
	Darolutamide	nmCRPCa
	Biclutamide, flutamide, cyproterone	CSPCa
	Apalutamide	mCSPCa, nmCRPCa
PI3K	Idelalisib	Chronic lymphocytic leukaemia, small lymphocytic leukaemia, follicular lymphoma
mTOR	Everolimus	HR+/HER- negative breast cancer, neuroendocrine tumours, renal cell carcinoma
	Sirolimus	Prevention of organ rejection following transplant
EGFR	Panitumumab	Metastatic colorectal cancer
	Cetuximab	Metastatic colorectal cancer, head and neck cancer in combination with radiotherapy
	Erlotinib, osemertinib, gefitinib, lapatinib	Non-small cell lung cancer
Nf-κB	Bortezomib	Multiple myeloma, mantle cell lymphoma
Bcl-2	Venetoclax	Chronic lymphocytic leukaemia, small lymphocytic leukaemia, follicular lymphoma, acute myeloid leukaemia,

## Data Availability

Not applicable.

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
