# Peer review of "Drivers of Radioresistance in Prostate Cancer"

_jcm, 2022, doi:10.3390/jcm11195637_

Round 1

Reviewer 1 Report

Interesting topic for this review paper.

Comprehensive review of resistance pathways.

I felt that the Conclusion is an anti climax

There is more scope to expand on targetable mechanisms of resistance 

No real mention of attempts to overcome resistance 

It would be topical to offer solutions to this challenge 

Author Response

Please attached response

Reviewer 2 Report

The topic of this study is of significant relevance in oncological practice.

Major points: 

It is important to explain the methods used for the study. Selection of studies? How have studies been selected.

As this is a journal of clinical medicine, I recommend to clearly separate the available studies into a) radiobiological studies and b) clinical studies.

For example: What are studies that show us that aniandrogen treatment combined with radiotherapy improves survival? Which studies show that castration resistant cancers are resistant to radiotherapy?

Minor point:

line 31 - androgen deprivation therapy is not a curative treatment as monotherapy

Author Response

Please see attached response

Round 2

Reviewer 2 Report

Thank you for the revisions.

There are a lot of clinical studies, including randomized studies, for antiandrogens in combination with radiotherapy. Furthermore, several clinical studies with new generation antiandrogens have also been published. They are routinely applied in clinical practice. Thus, the section on clinical applications need to be revised and these clinical studies mentioned and discussed.

In Table 1, several "currently approved inhibitors" have been listed. To avoid misunderstandings, please state which drugs have been approved for prostate cancer and who has approved these drugs.

Round 3

Reviewer 2 Report

The manuscript is acceptable for publication following the last revision.